# Attentional Control in Bilingualism: An Exploration of the Effects of Trait Anxiety and Rumination on Inhibition

**DOI:** 10.3390/bs9080089

**Published:** 2019-08-19

**Authors:** Julia Ouzia, Peter Bright, Roberto Filippi

**Affiliations:** 1Department of Psychology and Human Development, Institute of Education, University College London, Institute of Education, Department of Psychology and Human Development, London WC1E 6BT, UK; 2Multilanguage & Cognition lab (MULTAC), University College London, London WC1E 6BT, UK; 3Faculty of Science and Engineering, Department of Psychology, Anglia Ruskin University, Cambridge CB1 1PT, UK

**Keywords:** bilingualism, Attentional Control Theory, executive function, trait anxiety, rumination, inhibitory control, eye tracking

## Abstract

Bilingual individuals have been reported to show enhanced executive function in comparison to monolingual peers. However, the role of adverse emotional traits such as trait anxiety and rumination in bilingual cognitive control has not been established. Attentional Control Theory holds that anxiety disproportionately impacts processing efficiency (typically measured via reaction time) in comparison to accuracy (performance effectiveness). We administered eye tracking and behavioural measures of inhibition to young, healthy monolingual and highly proficient bilingual adults. We found that trait anxiety was a reliable risk factor for decreased inhibitory control accuracy in bilingual but not monolingual participants. These findings, therefore, indicate that adverse emotional traits may differentially modulate performance in monolingual and bilingual individuals, an interpretation which has implications both for ACT and future research on bilingual cognition.

## 1. Introduction

In an increasingly globalised world, in which over half the population is considered bilingual [1], the ability to communicate in more than one language offers a range of personal and professional advantages. Whether and how the processing of two or more languages in one mind may alter cognition has been the focus of a considerable body of research, and the argument that bilingualism offers genuine cognitive advantages has been increasingly challenged in recent years (see [2] for a comprehensive overview of the debate). Some empirical evidence suggests that, in comparison to monolinguals, bilingual individuals across the lifespan and from a range of linguistic backgrounds are faster and less affected by conflicting response demands when performing tasks measuring executive function (e.g., [3,4]). In particular, a bilingual advantage has been reported on measures of inhibition (e.g., [4,5]), attention shifting (e.g., [6,7]) and updating in working memory [4,8], although such claims are countered by evidence that observed advantages are typically small and statistically unreliable, particularly when considered in the context of publication bias towards reporting positive effects [9,10,11,12]. 

One interpretation of these effects derives from psycholinguistic evidence that bilinguals’ two languages are simultaneously activated at all times, even in unilingual contexts [13]. One of the most influential theoretical frameworks, the Inhibitory Control Model [14,15] (ICM), proposes that this unique form of language processing requires the active inhibition of one language in favour of producing the other (see [16] for an alternative explanation). According to the ICM, in order to resolve the competition between the two languages, cognitive control mechanisms are required. It is the additional cognitive effort associated with processing of two (or more) languages that is, therefore, thought to lead to enhancement in executive function [17]. A recent development of the ICM, the adaptive control hypothesis [18], further postulates that the kinds of control mechanisms used in bilingual speech production adapt according to the demands of an individual’s everyday interactional context, with an increased need to switch between two languages leading to a broader range of cognitive control advantages.

However, in the context of recent challenges to theory, concerns about methods and the appropriateness of statistical analyses employed in bilingual cognition research [2], the question of whether or not bilingualism is associated with enhanced cognitive abilities remains fiercely debated. One theme that has emerged is that if a bilingual advantage does exist, it may be task-specific or otherwise operate only across particular groups of participants [19,20]. Crucially, it may also be premature to speak of a universal bilingual *advantage* in non-verbal cognitive functioning in light of a recent report of a bilingual disadvantage in metacognition [21]. Nevertheless, given the prevalence of bilingualism and the implication this may have for professional practice of, for example, educators and clinicians, it remains an important endeavour for scientists to chart and understand the broad implications of multilanguage acquisition on cognition.

An important gap in the literature to date is that, to our knowledge, no studies have addressed the question of whether individual emotional states or traits may differentially affect bilinguals’ and monolinguals’ performance on tasks measuring executive functions. To our knowledge, there is only one study that has investigated emotion processing, specifically emotion regulation, in bilingual individuals [22]. Janus and Bialystok administered the Emotional Face *n*-Back task to 9-year-old monolingual and bilingual children. In this task, participants must indicate whether a letter has been shown on the previous screen (1-back) or on the screen before that (2-back), while faces displaying an angry, happy, or neutral expression are shown on both sides of the letter. The authors found that, whilst bilingual children performed the task more accurately overall and more slowly in the 2-back condition, the effects of emotional valence on reaction time did not differ across groups. They interpreted these findings as evidence that bilingual children may be at an advantage in terms of adjusting their behaviour to task demands but not in terms of emotion regulation. Whilst this study may add to our understanding of emotion processing in bilingual individuals, the focus was on cognitive performance of a task involving emotionally valanced stimuli rather than the effect of the emotional states or traits of the participants themselves. Anxiety and other mood disorders are among the most commonly occurring mental health problems, representing a substantial burden to the economy (e.g., [23]). The present study explores the effects of trait anxiety and rumination on inhibition, as conceptualised by Attentional Control Theory [24] (ACT), in young, healthy monolingual and bilingual adults. 

The ACT relies on the assumption that anxiety (in both clinical and non-clinical populations) adversely affects processing efficiency (typically inferred via reaction times) to a greater extent than performance effectiveness (i.e., accuracy) [25]. Specifically, in order to prevent anxiety from adversely affecting their performance, highly anxious individuals are thought to modulate the amount of effort they exert on difficult cognitive tasks, thus operating at a decreased level of efficiency in comparison to individuals with low levels of anxiety. The theory further assumes that there are two attentional systems [26,27]: one goal-directed (top-down) and the other stimulus-driven (bottom-up). Anxiety is thought to alter how these two attentional systems are balanced, with the presence of threatening stimuli decreasing goal-directed and increasing stimulus-driven attention. Eysenck and colleagues also argued that the challenges of maintaining goal-directed attention through inhibition and shifting should be most affected by anxiety, whereas storing information (updating) is not directly linked to attentional control and, thus, should not be associated with these effects as strongly and only be observable under particularly stressful conditions (although note that trait worry, a component of trait anxiety, has been related to updating [28,29]).

A substantial body of work has provided empirical support for the individual assumptions and hypotheses of the ACT. One method for testing the effects of anxiety on attentional control is the assessment of continuous overt visual attention via analysis of eye movements (saccades) [30,31] on the antisaccade task (note that the antisaccade task incorporates both pro- and antisaccade conditions) [32]. This task provides a measure of visual inhibition [33] because it incorporates an antisaccade condition in which the participant is required to produce a saccade to the opposite side of space from a visually presented stimulus [34,35,36]. Derakhshan, Ansari, Hansard, Shoker, and Eysenck [37] tested sixty-one healthy adults on two versions of the task, one featuring an oval stimulus (classic version) and the other neutral, happy, and angry faces. The study mainly focused on the latency of the first saccade made on each antisaccade trial, which is argued to be an indicator of processing efficiency (i.e., it is typically prolonged due to the requirement to inhibit a reflexive saccade to the stimulus). Furthermore, Derakhshan and colleagues assessed saccadic, as well as behavioural, accuracy (performance effectiveness) and corrective behaviours (correcting an erroneous saccade within the same trial). The latter, they argued, could be an indicator of compensatory strategies used in difficult (antisaccade) trials by high-anxiety participants. 

All participants completed the Trait Anxiety Scale of the State-Trait Anxiety Inventory [38], which is a well-established self-report measure assessing how individuals feel about themselves in general (Cronbach’s α = 90 [39]). The authors conducted a tertile split and only included those with the highest and the lowest trait anxiety scores in the analysis. In line with the assumptions of the ACT, the authors found that high-anxiety individuals showed reduced processing efficiency when the task was difficult (i.e., they produced longer antisaccade latencies), but did not differ from the low-anxiety group on any of the performance effectiveness measures (saccadic and behavioural accuracy, corrective behaviours), or on prosaccade performance. Furthermore, they found that the presence of threatening stimuli (angry faces) disproportionately affected processing efficiency in high-anxiety individuals, thereby supporting the ACT hypothesis that anxiety decreases goal-directed attention in favour of increased stimulus-driven attention.

In a later study, De Lissnyder, Derakhshan, De Raedt, and Koster [40] assessed these effects in a healthy population differentiated in terms of general depressive symptoms, as well as rumination. Rumination is a cognitive symptom of depression, which manifests itself in recurrent thoughts, contemplating the symptoms, causes, as well as implications of one’s depressive state [41]. This disposition to self-focus has previously been argued to be a key element of cognitive vulnerability associated with depression [42,43] De Lissnyder and colleagues [40] administered the self-report Ruminative Response Scale [44] in order to assess participants’ overall ruminative tendencies, as well as the two distinct subtypes of rumination, *reflective pondering* (the focus on problem solving; adaptive rumination) and *depressive brooding* (the focus on one’s negative mood; maladaptive rumination) [45]. They administered the classic antisaccade task, in order to assess inhibition, as well as a mixed (shifting) version of the task, in which the direction of the gaze is determined on a trial-by-trial basis by a cue displayed in the fixation period [46]. The authors found that two groups with high and low general depression did not differ in their performance of the antisaccade task. In contrast, the high-rumination group was found to display slower antisaccade latencies when compared to the low-rumination group, with depressive brooding being a predictor of antisaccade latencies in particular. Thus, the study of De Lissnyder and colleagues [40] replicated the findings reported by Derakhshan and colleagues [37], demonstrating that attentional control deficits are not only associated with high trait anxiety, but also with high levels of depressive brooding/maladaptive rumination. In line with previous research [47], rumination was not associated with deficits in shifting.

Research employing both behavioural methods, as well as neuroimaging, has provided further support for the ACT (see [48,49,50] for reviews). For example, state and trait anxiety, as well as chronic stress, have been found to predict reduced shifting abilities on a variety of tasks [46,51,52]. Furthermore, prefrontal response differences have been identified in neuroimaging and electrophysiological studies between low- and high-anxiety individuals in the absence of behavioural inhibition differences [53,54,55]. Therefore, there is substantial evidence in support of the ACT, deriving from studies using a variety of paradigms.

### The Current Study

The current investigation did not seek to address the bilingual executive function advantage per se, but rather sought to evaluate what this commonly postulated advantage may mean within the context of the hypotheses posed by the ACT. Informed by previous literature, we administered a classic version of the antisaccade task, as well as a behavioural measure of inhibition, the Simon task [56,57], to young, healthy monolingual and bilingual adults. Whether or not bilingualism is associated with differential oculomotor control abilities is unresolved given the sparse and conflicting evidence currently available [58]. With regards to the Simon task, Bialystok and colleagues [59] found evidence that, in the absence of consistent evidence for a behavioural advantage, activation in the dorsolateral prefrontal cortex was associated with faster reaction times in the Simon task only in monolinguals, whereas bilinguals were found to recruit resources in language processing areas of the brain alongside other regions in the left frontal hemisphere. The authors interpreted this to be evidence for the notion that the management of two language systems impacts non-verbal cognitive processing such that bilingual individuals recruit a more diverse network of cortical areas in the service of more efficient processing [60]. To date, there is a scarcity of studies testing the ACT in light of individual differences. There is some evidence suggesting that increased working memory capacity may serve as a protective mechanism against the adverse effects of anxiety on performance (e.g., [61,62]). Given that the ACT predicts that increased levels of anxiety are associated with a more dispersed allocation of attentional resources [24], it is reasonable to assume that individual differences in cognitive functioning, such as those reported by some studies comparing monolingual and bilingual individuals, will lead to differences in the effect of anxiety on inhibition [61]. We predicted that trait anxiety and rumination would not impact performance in either group on easy/congruent trials in either task, but that between-group performance would diverge on the more demanding incongruent conditions. The key question here was whether the commonly postulated inhibitory control differences between monolingual and bilingual individuals, behaviourally and/or on a neural level, would lead to differential effects of trait anxiety and rumination on cognitive performance.

## 2. Methods

### 2.1. Participants

Sixty-two young healthy adults, half of which were monolinguals and the other half bilinguals from a range of linguistic backgrounds (*n* = 16), were recruited for this study. One bilingual participant could not complete the eye tracking task due to technical issues. Therefore, thirty-one English monolinguals (*M_age_* = 22.3, *SD* = 3.7, range 18.3–34.4; 12 males) and a group of thirty bilinguals (*M_age_* = 25.3, *SD* = 4.5, range 19.6–38.3; 13 males) completed all elements of testing. Bilingual participants completed a language history questionnaire adapted from [62], which revealed that, overall, the group had high levels of English language proficiency (see Table 1).

None of the participants reported to have a history of mental health difficulties or neurological deficits. Participants with corrected-to-normal vision were asked to wear clean glasses during the eye tracking procedure, although a small minority of participants opted to not wear glasses as they usually only wore them for specific activities, such as driving. One participant wore contact lenses, but their gaze data did not appear to be affected by this and they were thus included in the analysis. 

### 2.2. Ethical Considerations

An ethics application for this study was submitted to the Anglia Ruskin University Department of Psychology Research Ethics Panel and approved by the Faculty of Science and Technology Research Ethics Panel, which confirmed that the methods reported here fully adhered to the *Code of Ethics and Conduct* outlined by the British Psychological Society [63].

### 2.3. Psychometric Materials

Measures of non-verbal reasoning and working memory were administered in order to evaluate whether the two groups were comparable with regard to general cognitive abilities. Emotional trait measures were employed and bilingual participants’ English Language proficiency was also assessed.

#### 2.3.1. Non-Verbal Reasoning: Raven’s Advanced Progressive Matrices (First Set)

In this test of nonverbal fluid intelligence, participants are presented with twelve trials. In each trial, they are shown an incomplete matrix of black and white abstract figures. Participants were asked to identify the missing piece from a selection of eight alternatives and to complete all 12 trials. Typically, participants completed this test within 10 min. None of the participants reached this time limit.

#### 2.3.2. Working Memory: Wechsler Adult Intelligence Scale (WAIS-IV) Digit Span Task

In this task, participants are asked to repeat sets of digits (with each set of sequences ranging from two to nine items) after oral presentation by the experimenter. During the first block (eight sets of two trials), they are asked to repeat the numbers in the same order; in the second round (seven sets of two trials), they are required to repeat the numbers in reverse sequential order. Each round is terminated once a participant has failed to correctly repeat both trials of one set, and a total score is calculated with a maximum of thirty points.

#### 2.3.3. Rumination, Reflective Pondering, and Depressive Brooding: Ruminative Response Scale (RRS) 

The RRS is a 22-item self-report measure of rumination that can be used to assess general ruminative tendencies, as well as the specific rumination sub-types, reflective pondering and depressive brooding [45]. Participants are asked to indicate what they generally think or do when they feel down, sad, or depressed on a scale of 1 (almost never) to 4 (almost always). Examples of general items are statements such as ‘Think about how alone you feel’ or ‘Think “Why can’t I get going?”’; reflective pondering examples are ‘Analyze recent events to try to understand why you are depressed’ or ‘Go someplace alone to think about your feelings’, and depressive brooding is assessed through items like ‘Think “What am I doing to deserve this?”’ or ‘Think about recent situation, wishing it had gone better’.

#### 2.3.4. Trait Anxiety: State-Trait Anxiety Inventory (STAI) 

The trait anxiety sub-test of the STAI consists of 20 statements which the participants should consider with regards to how they generally feel, with responses ranging from 1 (almost never) to 4 (almost always; e.g., ‘I feel rested’ or ‘I have disturbing thoughts’).

#### 2.3.5. Bilingual English Language Proficiency: Picture Vocabulary Subtest of the Bilingual Verbal Ability Tests (BVAT) 

Picture naming has successfully been used to evaluate bilingual individuals’ second language proficiency in previous research (e.g., [64,65,66,67]) and, therefore, a shortened version of the Picture Vocabulary subtest of the BVAT was employed for this purpose. In this subtest, participants are shown 58 images and need to either identify the image as a whole, part of the image, or an action displayed in the image. As all of the bilingual participants who took part in this research project were adults who had completed, or were in the process of completing, degree-level education in the UK at the time of testing, the first sixteen items on the Picture Vocabulary subtest were not used (sixteen points were added to participants’ scores) in order to limit the amount of time required for participating in the studies. 

### 2.4. Antisaccade Task

A classic version of the antisaccade task (containing pro- and antisaccade blocks) was programmed in E-Prime 2.0 Professional [68], according to the descriptions provided by Derakhshan and colleagues [37], and presented on the eye tracker via Tobii Studio. A six-point calibration was conducted in Tobii Studio [69] and a nine-point calibration was conducted in E-Prime before the task commenced. Both calibrations were successful for all participants. 

The task started with two practice blocks (one pro- and one antisaccade) containing six trials each. The experimental phase contained six blocks (three pro- and three antisaccade) with twenty trials each. Each block was preceded by a response-terminated instruction screen instructing the participant to either look towards the stimulus appearing on the screen or away from it, to the opposite side of the screen. The order in which blocks were presented was randomised across participants. Within each block, trials were presented consecutively without breaks, each starting with a fixation display, which lasted until the participant fixated the cross at the centre of the screen for 1000 ms. This was done to ensure that the participant returned to the centre of the screen at the end of each trial and to identify any technical errors with the equipment. After this, a stimulus appeared either on the left or right side of the screen (to an equal amount within each block) for a period of 600 ms. See Figure 1 for an illustration of pro- and antisaccade trials. E-Prime Extensions for Tobii [70] was used to save gaze data for analysis.

The stimuli displayed in the task were a white oval shape, as well as a fixation cross, created in Adobe Photoshop CS6 (dimensions informed by Derakhshan and colleagues, [37]).

#### Eye Tracking Equipment

For the collection of eye movements, the Tobii 1750 eye tracking system was used [68], which has a 17-inch TFT-LCD monitor and a resolution of 1280 × 1024. The system uses a version of the Pupil Centre Corneal Reflection method [71], in which infrared light is directed at the cornea and pupils of both eyes, creating a reflection that is captured by a sensor. The sampling rate is approximately 50 HZ (20 ms), with an accuracy of 0.50 and a latency of 35 ms. The system allows a head-movement of 30 × 16 × 20 cm (width, height, and distance, respectively) and has a compensation error of <10. An adjustable headrest was placed 60 cm away from the screen (70 cm away from the sensors). 

### 2.5. Simon Task

The Simon task was programmed according to instructions provided by Bialystok and colleagues [4] in E-Prime 2.0 Professional [68]. Each trial began with a fixation cross displayed in the centre of the screen for a period of 800 ms, followed by a blank interval presented for 250 ms. Following this, a screen in which a blue or a red square appeared on either the left or right side of the screen was presented (equally and randomly distributed across the task), and remained on the screen until the participant responded or for a maximum of 1000 ms if no response was made. There was a 500 ms blank interval between trials. The task began with a practice phase that terminated after eight correct responses were made consecutively. The experimental phase consisted of twenty-eight trials (blue/right, blue/left, red/right, red/left for seven trials each). Participants were instructed that they should press the left key (‘a’) when a blue square appeared on the screen and the right key (‘l’) when a red square appeared, and were asked to do so as quickly and accurately as possible.

### 2.6. Procedure

Half of the monolingual and bilingual participants completed the trait anxiety subtest of the STAI and the RRS before the behavioural measures were administered, whereas the other half completed them afterwards. Eye tracking was conducted in a room without windows and lighting conditions were kept constant across participants. The basic principles of the task were explained to the participant before eye tracking began and they were told to read the instruction screens carefully each time as those contained the appropriate gaze direction. After calibration, the practice blocks were presented, starting with the prosaccade block. The rules of the task were then reiterated, the participant was reminded to take breaks between blocks if necessary, and to keep their head as still as possible on the rest. Following eye tracking, participants completed the Simon task. Inclusive of information and debrief provision, each session lasted approximately 45 min. All participants were entered into a raffle for a £25 voucher.

## 3. Results

### 3.1. Gaze Data Preparation

The raw eye movement data were analysed with respect to average performance across both eyes using Microsoft Excel. Blinks were eliminated and point-by-point velocities and amplitudes were calculated using formulae recommended by Salvucci and Goldberg [72]. 

The requirements for a saccade were informed by Derakhshan and colleagues [37] (velocity of >300/s; amplitude of >30/s; minimum onset time of 83 ms). Additionally, saccades had to be followed by eye movements in the same direction; otherwise, qualifying saccades were considered to be noise and thus disregarded. Trials that did not feature saccades or were not recorded due to eye-tracking failure were also excluded from the data, resulting in a slight drop in the average number of trials per condition and group (monolingual prosaccade: *M* = 58.52, *SD* = 4.46; monolingual antisaccade: *M* = 57.52, *SD* = 2.76; bilingual prosaccade: *M* = 58.70, *SD* = 1.56; bilingual antisaccade: *M* = 56.52, *SD* = 2.08). The accuracy data presented here are percentages of qualifying trials; independent samples *t*-tests revealed that the number of completed trials across groups in pro- and antisaccade conditions was comparable (*p*s > 0.05). Accounting for completed trial rate in the antisaccade analyses reported below did not alter the findings. The reaction time data reported here are based on correctly performed trials only.

Data were parsed with regards to three main dependent variables, separately for pro- and antisaccade trials: (1) latency of first correct saccade, (2) accuracy, and (3) percentage of corrective saccades. An accurate saccade was defined as a saccade moving in the direction required on each trial, concluding in a fixation on the location. Trials in which corrective saccades were made (the correction of an erroneous saccade) were classified as inaccurate.

### 3.2. Outliers

Data from three bilingual participants who completed all blocks according to prosaccade instructions were removed. Furthermore, two monolingual participants completed one block of prosaccade trials according to antisaccade task instructions (each). These blocks were removed from their respective data sets. No further corrections were made.

### 3.3. Group Differences on Psychometric Measures and Age

A One-Way ANOVA, evaluating between-group performances on controlling measures, as well as age, revealed that the two groups performed comparably on all controlling measures, but that the bilingual group was significantly older than the monolingual group (see Table 2).

### 3.4. Antisaccade Task

A series of 2 * 2 mixed ANOVAs, comparing the two groups’ latency of first correct saccade, accuracy, and percentage of corrective saccades on pro- and antisaccade trials, revealed that the groups performed the task comparably (*p*s > 0.29). Antisaccade blocks were associated with longer first saccade latencies, *F*(1, 56) = 392.18, *p* < 0.001, *η*^2^ = 0.88, lower levels of accuracy, *F*(1, 56) = 73.38, *p* < 0.001, *η*^2^ = 0.57, as well as higher percentages of corrective saccades, *F*(1, 56) = 54.16, *p* < 0.001, *η*^2^ = 0.49. No interaction effects were detected (*p*s > 0.36). Table 3 summarises monolinguals’ and bilinguals’ performance.

Accounting for group differences in age did not alter the results reported here and bilinguals’ BVAT scores were not found to predict any of the dependent variables (*p*s > 0.08).

### 3.5. Antisaccade Task: Trait Anxiety and Rumination

The effects of trait anxiety and rumination on pro- and antisaccade trial performance were assessed in individual linear regressions for each group and the regression coefficients were then compared across groups (as outlined by [73]). Where the rumination regression coefficients were significantly different, models assessing the effects of reflective pondering and depressive brooding were considered. Outliers were evaluated using Cook’s distance [74] and removed where necessary. The analyses concerning prosaccade trials are reported in Appendix A, none of which yielded significant findings.

#### 3.5.1. Trait Anxiety

Trait anxiety significantly predicted accuracy on antisaccade trials in bilinguals, *β* = −0.41, *t*(25) = −2.23, *p* = 0.035, but not monolinguals, *β* = 0.06, *t*(29) = 0.30, *p* = 0.769. The difference between the two regression coefficients was significant, *t*(54) = −2.12, *p* = 0.039. 

The effect of trait anxiety on percentage of corrective saccades approached significance in bilinguals, *β* = 0.37, *t*(25) = 1.98, *p* = 0.059, and was non-significant in the monolingual group, predicting an effect in the opposite direction, *β* = −0.06, *t*(29) = −0.33, *p* = 0.743. The difference between the coefficients was non-significant, *t*(54) = 1.74, *p* = 0.087. However, the removal of one monolingual outlier (Cook’s *D* = 1.04) rendered the difference in slope significant (monolingual slope: *β* = −0.31, *t*(28) = −1.73, *p* = 0.095; difference: *t*(53) = 2.65, *p* = 0.011). 

Therefore, trait anxiety negatively affected bilinguals’ accuracy and predicted a higher number of corrective saccades on antisaccade trials and these effects were significantly different from those in the monolingual group (see Figure 2 for an illustration of all effects).

No other effects or group differences corresponding to levels of trait anxiety were found (see Table 4 for regression coefficients and *t*-statistics). Post-hoc power analyses were conducted on the effect of trait anxiety on antisaccade latencies reported in previous literature, e.g., [37,40] in monolinguals, *β* < −0.06, *t*(29) = −0.32, *p* = 0.755, revealed an observed power of 0.06. The importance of this finding will be addressed in the discussion.

#### 3.5.2. Rumination, Reflective Pondering and Depressive Brooding

Rumination was found to significantly predict bilinguals’ antisaccade accuracy, *β* = −0.39, *t*(25) = −2.13, *p* = 0.043. Further analyses revealed that depressive brooding significantly contributed to this effect, *β* = −0.54, *t*(24) = −2.68, *p* = 0.013, but not reflective pondering, *β* < −0.01, *t*(24) = −0.01, *p* = 0.989. The model accounted for 28.9% of the variance, *F*(1, 24) = 4.87, *p* = 0.017. In monolinguals, all effects were non-significant (rumination: *β* = 0.03, *t*(29) = 0.15, *p* = 0.879; depressive brooding: *β* = 0.17, *t*(29) = 0.69, *p* = 0.495; reflective pondering: *β* = −0.20, *t*(29) = −0.84, *p* = 0.410). The differences between the effects of rumination and depressive brooding in the two groups were significant, respectively, *t*(52) = −2.05, *p* =0.045; *t*(52) = −2.70, *p* = 0.009. Therefore, rumination as a whole and, specifically, depressive brooding predicted significantly reduced antisaccade accuracy in bilingual individuals but not in monolinguals (see Table 5 for regression coefficients and *t*-statistics of all rumination effects).

### 3.6. Simon Task

We also ran two 2 * 2 mixed ANOVAs comparing monolinguals’ and bilinguals’ accuracy and reaction times on congruent and incongruent trials of the Simon Task. This revealed that the groups performed the task comparably (*p*s > 0.08). Incongruent trials were associated with slower performance, *F*(1, 56) = 44.81, *p* < 0.001, *η*^2^ = 0.45, as well as lower levels of accuracy, *F*(1, 56) = 16.85, *p* < 0.001, *η*^2^ = 0.23. No interaction effects were detected (*p*s > 0.13). Table 6 summarises monolinguals’ and bilinguals’ performance.

Accounting for group differences in age revealed that, overall, monolinguals performed the task faster than bilinguals, *F*(1, 55) = 4.25, *p* = 0.044, *η*^2^ = 0.07. However, the Simon Effect was comparable across groups (*p* = 0.31), indicating similar behavioural inhibition abilities. In the bilingual group, English vocabulary knowledge measured by the Picture Vocabulary subtest of the BVAT was found to significantly predict reaction times on incongruent trials of the task, *r*(25) = 0.39, *p* = 0.042, all other correlations were non-significant (*p*s > 0.30).

### 3.7. Simon Task: Trait Anxiety

The only notable effect detected when running the same models on Simon task data concerned the relationship between trait anxiety and accuracy on incongruent trials of the Simon task (see Appendix B for a full report of these analyses). Whilst the predictions were non-significant in both bilinguals, *β* = −0.32, *t*(25) = −1.67, *p* = 0.107, and monolinguals, *β* = 0.24, *t*(29) = 1.33, *p* = 0.194, the directionality of effects differed across groups, with higher levels of trait anxiety predicting lower levels of accuracy in the bilingual and higher levels in the monolingual group. The comparison of regression coefficients revealed that this difference in effects approached significance, *t*(54) = −1.94, *p* = 0.058. Three monolingual participants performed the task considerably less accurately than other participants in the monolingual group (Figure 3), leading to a negative skew of the data. We, therefore, ran the same analyses excluding these participants. This yielded similar results but a significant difference in slopes (bilinguals: *β* = −0.32, *t*(25) = −1.67, *p* = 0.107; monolinguals: *β* = 0.23, *t*(26) = 1.20, *p* = 0.241, difference: *t*(51) = −2.05, *p* = 0.046).

## 4. Discussion

The main objective of this investigation was to evaluate whether monolingual and bilingual individuals differed with regard to the effects of adverse emotional traits on the performance of two measures thought to tap into inhibition. Informed by previous literature, a combined version of the classic pro- and antisaccade tasks, as well as the Simon Task, were employed. The effects of trait anxiety and rumination, as well as the rumination types, reflective pondering and depressive brooding on performance effectiveness and processing efficiency were assessed within both groups and compared across groups.

The main hypothesis of this study, speculating that bilinguals may be differentially affected by adverse emotional states and traits, was confirmed. The analyses revealed that trait anxiety predicted lower levels of performance effectiveness on the most difficult trials of both tasks only in bilingual participants. Remarkably, this effect was found on all variables of performance effectiveness, including saccadic and behavioural accuracy, as well as corrective behaviours. Significant effects of rumination were only found with regard to saccadic accuracy and, in line with findings reported by De Lissnyder and colleagues [40], depressive brooding but not reflective pondering was found to significantly predict performance. It has previously been noted that the focus on one’s negative mood, which is a characteristic specific to depressive brooding, may lead to attentional inflexibility and, thus, implicate inhibitory resources more so than reflective pondering [47].

No effects of the emotional variables on processing efficiency were detected in either group. It is important to note here, however, that our study took a different approach to others by focusing specifically on the effects of adverse emotional traits on cognitive performance. Published ACT studies have typically divided participants into low- and high-anxiety individuals (e.g., [37]) or high- and low ruminators [40], whilst in this investigation, the effects of adverse emotional traits were assessed on a continuum. Therefore, it is possible that the main assumption of the ACT relevant here, that adverse emotions effect processing efficiency as measured by first antisaccade latencies, can only be observed in a group that scores on the higher end of the spectrum on the emotional measures administered. This interpretation is in line with recent findings suggesting that correlations between performance and anxiety as a continuous variable are not always found, even when high- and low-anxiety individuals differ [75]. Testing this hypothesis was not possible given the sample size constraints of this study and we thus encourage a replication involving a larger pool of participants.

If bilingual individuals rely on inhibitory mechanisms when controlling their two languages, as hypothesised by the ICM [14,15], it is possible that these mechanisms become increasingly efficient over time. In turn, bilingual speakers may have less experience at exerting effort over these mechanisms compared to monolinguals. According to the ACT, anxiety should not have an impact on performance effectiveness (as measured by accuracy) but it should have it on processing efficiency (as indicated by response time). Whereas effectiveness is affected by adverse emotions, efficiency is modulated in order to compensate for these effects. Therefore, performance may still reach the same level, but under a condition of increased effort. If bilinguals do not use effort to modulate inhibitory mechanisms in everyday life, it is plausible that they will not make use of the protective functions of effort when their cognitive functioning is subject to the influence of adverse emotions. However, this interpretation is speculative and at odds with the literature challenging the bilingual advantage, as well as a recent argument that bilingual language control may not rely on executive function beyond the initial stages of second language learning [76]. Therefore, the evaluation of the differences reported here demands further investigation, for example, in studies employing a broader range of tests of executive function.

With regard to overall task performance, the two groups did not differ on the antisaccade task, which is in line with findings reported by Bialystok, Craik, and Ryan [77]. Previous research suggests that bilinguals’ level of second language proficiency is a predictor of their cognitive performance (e.g., [8,78,79], although note [80,81,82,83], who reported contradictory findings). Thus, the finding that bilinguals’ proficiency did not significantly predict performance in the antisaccade task, whilst it did on incongruent trials of the Simon Task, supports the notion proposed by Bialystok and colleagues [77] that the eye tracking version of the task detects very early processing effects that may not be subject to the bilingual advantage in inhibitory control. In other words, language ability does not appear to predict performance in these early attentional markers. With regard to the Simon Task, similarly to earlier research [84], no group differences in accuracy or the Simon Effect were detected in this sample of young healthy adults.

### Future Directions

Following on from the methodological considerations made above, it will be important for future research to further consider the effects reported in this study in groups of high- and low- anxiety monolinguals and bilinguals. Based on the findings reported here, as well as previous research addressing the assumptions and hypotheses of the ACT, it is possible that these effects are confined to highly anxious bilinguals and will become more pronounced as a result. Furthermore, it will be important to evaluate whether or not processing efficiency is affected differently by anxiety in bilingual individuals. If the current interpretation of these results is correct, i.e., bilinguals do not modulate effort in order to compensate for adverse emotional effects on performance, a high-anxiety bilingual group should not differ from a low-anxiety group with regard to processing efficiency. Alternatively, results could suggest that the effect of trait anxiety and rumination on processing efficiency is reduced in bilinguals, compared to monolinguals. The same pattern of results should emerge in a high-rumination group of bilinguals. However, considering that regression coefficients regarding effects of adverse emotions on processing efficiency did not significantly differ between groups in this study, it is possible that bilinguals experience a more widespread disadvantage in dealing with adverse emotions.

Recent research conducted by Berggen and Derakshan [75] suggests that anxiety may implicate stimulus–response competition in particular, as opposed to stimulus–stimulus competition. Notably, to our knowledge, this was the first study evaluating the impact of anxiety on distractor cost (i.e., the difference between congruent and incongruent trials), as opposed to treating congruency as an independent variable with two levels, as is common in the ACT literature, e.g., [37,40,85]. Given current challenges to how inhibition has been conceptualised by the literature to date, e.g., [81,86], it will be important for future research to systematically evaluate the relationship between adverse emotional traits and different types of inhibition in both monolingual and bilingual individuals. 

The question of whether bilingual individuals process threat-related stimuli similarly to monolinguals should also be addressed in future research. Based on the findings reported here, we are hesitant to offer any firm predictions, considering that past research has extensively evaluated a bilingual advantage in inhibiting the presence of task-irrelevant visual stimuli, e.g., [87,88]. Research from our lab (see Ouzia and Filippi [89] for further details) suggests that the relationship between trait anxiety/depressive brooding and sentence comprehension accuracy in the presence of auditory distractors featuring adverse emotions (crying) differs among monolinguals and bilinguals. Specifically, it appears to suggest threat-avoidance in monolinguals, with the presence of a distractor being associated with more accurate performance, whilst bilinguals exhibit attentional bias (i.e., the distractor is associated with a decrease in accuracy). This indicates that, depending on the presentation of the threat-related stimulus, bilinguals may be faced with either an advantage or a disadvantage.

We suggest that evaluating the mechanisms with which adverse emotions affect cognitive functioning in bilingualism will be of great importance for theory, research, and applied work with bilingual individuals. Whilst it may seem counterintuitive at first, bilinguals’ reliance on inhibitory control mechanisms in everyday-life language processing may not always lead to observable advantages, but disadvantages as well. The notion that bilingualism may affect the ways in which individuals are able to exert additional cognitive effort when demanded by internal processes not directly linked to bilingual cognition, such as anxiety and rumination, will require further enquiry. In light of research suggesting that adverse effects of anxiety and how they are dealt with cognitively can impact motivational levels in learning [90] and also, the inherent importance of this research for understanding cognition in clinical populations [91], this line of inquiry delivers a promising direction for research on bilingualism.

## 5. Conclusions

In conclusion, the study reported here offers novel insight into how adverse emotions may affect cognition differentially in monolinguals and bilinguals. It appears that the increased demand for engagement of inhibitory control in bilingualism may render bilinguals more vulnerable to these effects. Future research should incorporate additional measures of cognitive control, larger sample sizes and a wider distribution of trait anxiety scores to confirm and better understand how the impact of emotional state on cognitive performance is modulated by processes associated with multilanguage acquisition.

## Figures and Tables

**Figure 1 behavsci-09-00089-f001:**
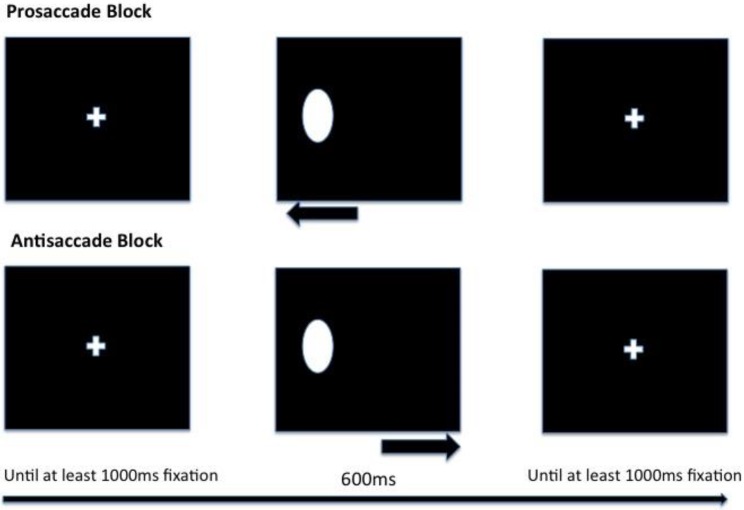
Presentation order of trials in pro- and antisaccade blocks with timings.

**Figure 2 behavsci-09-00089-f002:**
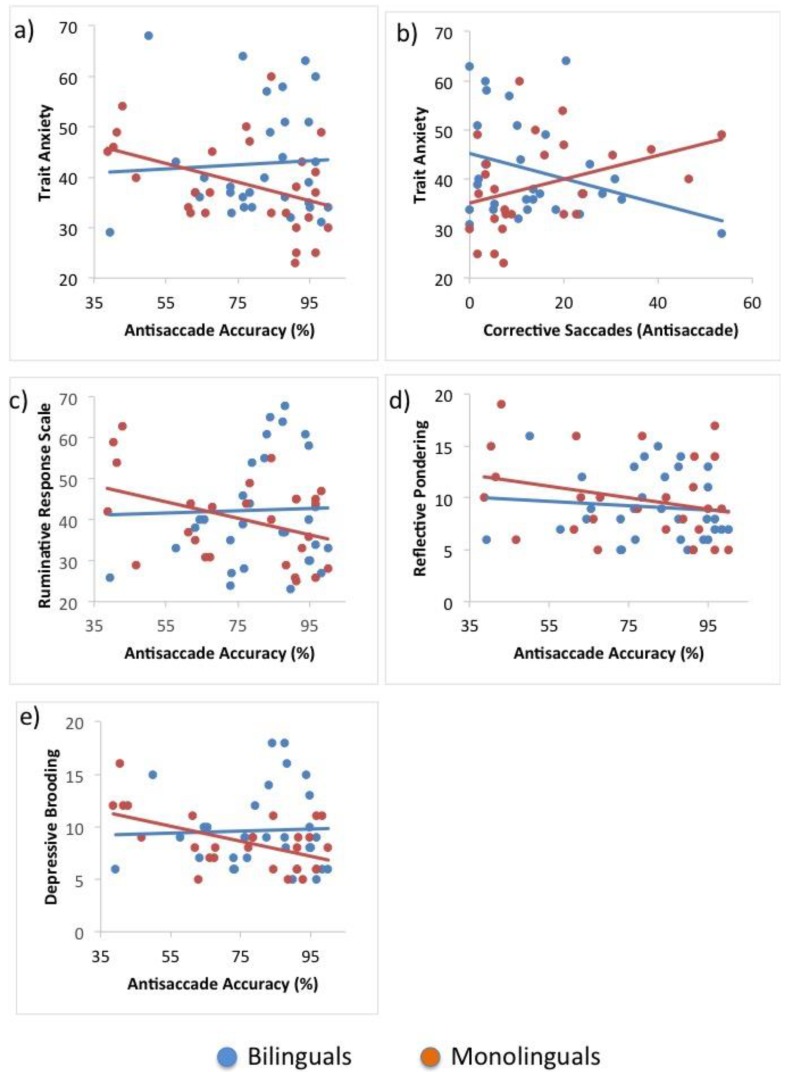
The effects of (**a**) trait anxiety on antisaccade accuracy; (**b**) trait anxiety on percentage of corrective saccades; (**c**) Ruminative Response Scale (RRS) score on antisaccade accuracy; (**d**) reflective pondering on antisaccade accuracy; and (**e**) depressive brooding on antisaccade accuracy in monolinguals and bilinguals.

**Figure 3 behavsci-09-00089-f003:**
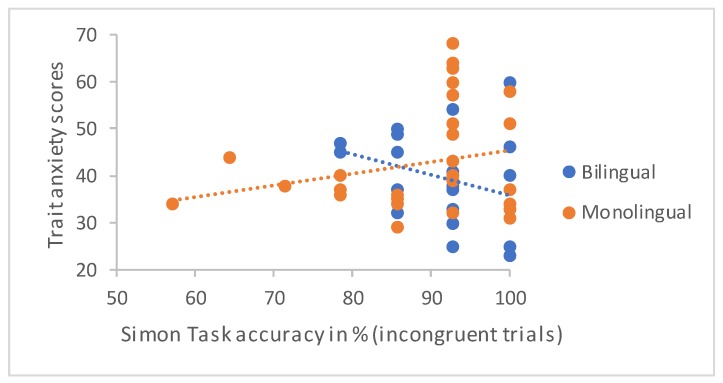
Effects of trait anxiety on accuracy on incongruent trials of the Simon Task in monolinguals and bilinguals.

**Table 1 behavsci-09-00089-t001:** Bilingual participants’ levels of self-rated English Language proficiency.

Linguistic background	First language	Bulgarian (*n* = 1)
		Creole (*n* = 1)
		Dutch (*n* = 2)
		Farsi (*n* = 1)
		French (*n* = 1)
		German (*n* = 2)
		Hindi (*n* = 1)
		Hungarian (*n* = 1)
		Italian (*n* = 2)
		Lithuanian (*n* = 1)
		Malayalam (*n* = 1)
		Polish (*n* = 7)
		Portuguese (*n* = 2)
		Romanian (*n* = 2)
		Sinhalese (*n* = 1)
		English (*n* = 4)
	Second language	Afrikaans (*n* = 1)
		English (*n* = 25)
		Frisian (*n* = 1)
		Greek (*n* = 1)
		Gujarati (*n* = 1)
		Twi (*n* = 1)
Other linguistic background information	Third language	English (*n* = 1)
	Age of first exposure	Birth-6 years (*n* = 14)
		7–12 years (*n* = 9)
		teenage years (*n* = 7)
	Time spent in the UK	0–5 years (*n* = 15)
		5–10 years (*n* = 9)
		10+ years (*n* = 6)
	Switch	rarely (*n* = 14)
		sometimes (*n* = 14)
		frequently (*n* = 2)
Self-rated proficiency (1–6)	Reading	*M* = 5.1; *SD* = 0.7
	Writing	*M* = 4.6; *SD* = 0.9
	Speaking	*M* = 4.9; *SD* = 0.8
	Listening	*M* = 5.2; *SD* = 0.7

**Table 2 behavsci-09-00089-t002:** Group means for age, working memory, non-verbal reasoning, trait anxiety, and rumination (standard deviations in brackets).

Variable	Monolinguals (*n* = 31)	Bilinguals (*n* = 27)	*F*-Statistic	
			*F*	*p*
**Age**	22.27 (3.69)	25.56 (4.69)	8.86	0.004
**Working Memory (max. 30)**	17.97 (4.85)	16.00 (3.84)	2.87	0.096
**Non-verbal Reasoning (max. 12)**	9.94 (1.65)	10.33 (1.57)	0.88	0.353
**Trait Anxiety (max. 80)**	42.68 (10.96)	38.85 (9.27)	2.03	0.160
**Rumination (max. 88)**	42.29 (14.33)	40.22 (10.43)	0.39	0.538
**Reflective Pondering (max. 20)**	9.13 (3.26)	9.96 (4.13)	0.74	0.394
**Depressive Brooding (max. 20)**	9.65 (3.72)	8.56 (2.72)	1.58	0.214

**Table 3 behavsci-09-00089-t003:** Group means of antisaccade task performance (standard deviations in brackets).

	Monolinguals		Bilinguals	
	Prosaccade	Antisaccade	Prosaccade	Antisaccade
**Latency of first correct saccade (ms)**	381 (23)	446 (29)	386 (24)	446 (30)
**Accuracy (%)**	98.27 (2.34)	80.65 (14.94)	97.77 (4.08)	75.91 (20.44)
**Corrective saccades (%)**	1.17 (1.93)	14.54 (13.43)	1.55 (2.40)	15.02 (14.09)

**Table 4 behavsci-09-00089-t004:** Regression coefficients and *t*-statistics of the relationship between trait anxiety and the dependent variables.

Dependent Variable	Monolinguals			Bilinguals			Difference
	*B*	*SE B*	*β*	*B*	*SE B*	*β*	
**First saccade latency**	−0.15	0.49	−0.06	0.09	0.64	0.03	*t*(54) = 0.30, *p* = 0.766
**Accuracy**	0.08	0.25	0.06	−0.90	0.40	−0.41 *	*t*(54) = −2.12, *p* = 0.039
**Percentage of corrective saccades**	−0.38	0.22	−0.31	0.56	0.28	0.37	*t*(53) = 2.65, *p* = 0.011

*p* < 0.05, * *p* < 0.01, *** *p* < 0.001.

**Table 5 behavsci-09-00089-t005:** Regression coefficients and t-statistics of the relationship between rumination, reflective pondering, and depressive brooding and the dependent variables.

	Dependent Variable	Monolinguals			Bilinguals			Difference
		*B*	*SE B*	*β*	*B*	*SE B*	*β*	
**Rumination**	**First saccade latency**	−0.15	0.37	−0.08	0.42	0.56	0.15	*t*(54) = 0.85, *p* = 0.397
	**Accuracy**	0.03	0.19	0.03	−0.77	0.36	−0.39 *	*t*(54) = −2.05, *p* = 0.045
	**Percentage of corrective saccades**	−0.07	0.17	−0.07	0.39	0.26	0.29	*t*(54) = 1.47, *p* = 0.146
**Reflective pondering**	**First saccade latency**	-	-	-	-	-	-	-
	**Accuracy**	−0.92	1.10	−0.20	−0.01	0.99	<−0.01	*t*(52) = 0.60, *p* = 0.549
	**Percentage of corrective saccades**	-	-	-	-	-	-	-
**Depressive brooding**	**First saccade latency**	-	-	-	-	-	-	-
	**Accuracy**	0.67	0.96	0.17	−4.02	1.50	−0.54 **	*t*(52) = −2.70, *p* = 0.009
	**Percentage of corrective saccades**	-	-	-	-	-	-	-

*p* < 0.05, * *p* < 0.01, *** *p* < 0.001.

**Table 6 behavsci-09-00089-t006:** Group means of Simon task performance (standard deviations in brackets).

	Monolinguals		Bilinguals	
	Congruent	Incongruent	Congruent	Incongruent
**Reaction time (ms)**	369 (48)	406 (51)	397 (64)	428 (64)
**Accuracy (%)**	96.54 (5.79)	88.94 (10.57)	96.56 (6.38)	93.12 (6.71)

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
