# Peer review of "Attentional Control in Bilingualism: An Exploration of the Effects of Trait Anxiety and Rumination on Inhibition"

_behavsci, 2019, doi:10.3390/bs9080089_

Round 1

Reviewer 1 Report

This is an interesting and innovative study exploring the relationships between inhibition, bilingualism, and trait anxiety/rumination.  To the extent that antisaccade trials and the incongruent trials in a standard Simon are valid measures of general inhibitory control, there was no evidence for a bilingual advantage in inhibitory control.  This outcome is not surprising.  Likewise, there were no differences between monolinguals and bilinguals on mean levels of trait anxiety or rumination. 

The only statistically significant difference between the language groups was the correlation between trait anxiety and antisaccade performance for bilinguals, but not for monolinguals.  More specifically, for bilinguals, as trait anxiety (or rumination) increases antisaccade accuracy decreases, whereas for monolinguals trait anxiety (or rumination) doesn’t matter. 

Two issues follow.  How should the results be interpreted and what level of confidence can be placed on the evidence? 

1.      Interpretation:  Trait anxiety has the potential to disrupt goal-directed action.  Disruption can be avoided by effective inhibition of irrelevant distracting information.  The high anxiety monolinguals in this study were apparently successful, but the bilinguals are not.  Why not?  Here’s my understanding or the argument, but I admit I am drawing some inferences that perhaps were not implied.  If so, the authors’ interpretation simply needs to be clarified.  The authors make the point that with continued practice inhibiting the non-target language may become automated (see Paap’s controlled dose hypothesis for a specific version of this hypothesis, 2018, in DeHouwer & Oretega’s Cambridge Handbook of Bilingualism).  If inhibition in bilingual language control is automated, then it is not enhancing or maintaining  general inhibitory control and, consequently, high anxiety hurts bilingual performance and results in the significant correlation.  But this does not explain why trait anxiety doesn’t matter for monolinguals.  The underlying logic of the interpretation needs to be clarified.   

2.     Quality of Evidence:  After losses due to eye tracking technical issues and failures to follow instructions the antisaccade analyses are based on 28 participants in each group.  Any between-group comparisons based on 28 in each group will not, of course, have desirable levels of power.   Perhaps more important, the significant correlation between trait anxiety and anti-saccade accuracy could be noise or could be due to some third variable that covaries with trait anxiety in this sample.  The likelihood of this pattern of results reproducing in a replication is pretty much a guess.  With respect to the Simon task the correlations between trait anxiety and accuracy on the incongruent trials were not significant for either language group.  However, the direction is positive for monolinguals (as anxiety increases Simon accuracy increases) and negative for bilinguals (as anxiety increases Simon accuracy decreases).   Directly comparing the slopes yields a “marginally significant”, p = .058, difference.   Inspection of Figure 3 raises another potential problem.  One to three (depending on the selected criterion) of the monolinguals could be considered outliers in terms of Simon accuracy.  To illustrate, one monolingual barely performed above chance (about 56% correct) and had a fairly low trait anxiety score.  This low level of performance suggests that this participant was not able to do the task (color blind?), did not understand and follow the instructions, or simply was not trying.  Eliminating this participant would probably render the “marginally significant” differences in slope even more marginal. 

Other comments

p. 1.  “a bilingual advantage has been reported on measures of inhibition, attention shifting, and updating in working memory”.  Although this is true, it is very misleading to readers who are not closely following the bilingual advantage in EF debate.  Several recent meta-analyses converge on the outcome that the bilingual-advantage effect size is very small and that when corrected for publication bias is not distinguishable from zero. 

Lehtonen et al. (2018) – 6 EF domains including inhibition, attention, shifting, & WM

Paap (2019; Schwieter’s Handbook of Neuroscience of Multilingualism) – inhibition & switching

Donnelly et al. (2019, PB&R) – inhibition

Von Bastian et al. (2018, Psychonomics) – WM

p. 4.  “prefrontal response differences have been identified in neuroimaging and electrophysiological studies between low- and high anxious individuals in the absence of behavioural inhibition differences, suggesting that high-anxious individuals need to exert more effort over inhibition and thus, are less efficient than those with low anxiety.”  There is a leap in the logic leading to this conclusion that escapes me.  Might the missing steps be filled in? 

p. 4.  Background information on the bilingual is quite sparse.  What non-English languages did they speak, which language was dominant, did they switch very often, etc.?

p. 5.  It is interesting that the authors treat WM (especially the backward span) as a control variable that should be matched across the two language groups rather than a component of EF that might yield a bilingual advantage. 

p. 6.  “either look towards or away from the stimulus appearing on the screen”  For antisaccade trials this seems somewhat vague.  From the participant’s point of view:  How far “away” should I look?  I suppose most participants would interpret this instruction to mean an equivalent distance on the opposite side from the ellipse. 

p. 7.  “The experimental phase consisted of twenty-eight trials….”  Why so few?  This is not typical for studies testing adults. 

p. 8.  “Monolingual antisaccade: M = 57.52…”  What are these units?

Second p. 1.  “The measure of interest on the Simon task was behavioral accuracy on difficult/incongruent trials given our goal of evaluating whether the effects of adverse emotional traits reported above could be observed in a behavioural measure of inhibition.”  The reasons behind this choice should be made explicit. 

1.     Why accuracy?  The vast majority of studies use RT because accuracy in young adults is usually near ceiling.   I think this choice may have something to do with the distinction between “efficiency” and “effectiveness”, but if so that should be clarified.  Furthermore, even if the underlying theory is more interested in “effectiveness” why would an analysis of RT (efficiency) not be of interest too? 

2.     Why mean RT on incongruent trials rather than interference scores?  Researchers often analyze both global RT (as a measure of monitoring) and interference scores (incongruent minus congruent) as a measure of inhibition.  Why this choice? 

Second p. 2.  “The main objective of this investigation was to evaluate whether monolingual and bilingual individuals differed with regard to the effects of adverse emotional traits on cognitive functioning, specifically inhibition.”  The implicit assumption here is that there is such a thing as domain-general inhibitory control.  Several researchers have suggested that inhibition is task specific (see discussion in Paap et al., 2019, JML) and some have recommended that we stop talking about inhibition (Rey-Mermet, et al., 2018, JEP:LMC).   The authors may want to avoid this quagmire.  On the other hand, it is sometimes good to acknowledge a brewing controversy that targets a basic assumption of the study design. 

Second p. 2.  “…this effect may occur because a continual focus on depressive thoughts may particularly implicate inhibitory resources”.  I don’t understand this. 

Second p. 2.  “If bilingual individuals rely on inhibitory mechanisms when controlling their two languages at all times, it is possible that these mechanisms become increasingly automated and efficient over time.  In turn, bilingual speakers may have less experience at exerting effort over these mechanism (e.g., when demanded by their emotional traits) compared to monolinguals.”

Although it is not clear, the authors appear to make the argument that the inhibitory control used in bilingual language control is the same as that used in nonverbal tasks such that when inhibitory control becomes automated in language switching this can carry over to the antisaccade or Simon task.  This would not be in the spirit of the literature on automaticity in skill acquisition.  What becomes automatic are action sequences specific to the task that receives extensive practice.  This would have no effect on domain-general and top-down inhibitory control.  I apologize if I have completely misinterpreted what the authors are saying in this quote.

Second p. 2.  “If bilinguals do not use effort to modulate inhibitory mechanisms in everyday life, it is plausible that they will not make use of protective functions of effort when their cognitive functioning is subject to the influence of adverse emotions.”  I am not sure what this sentence means.  Does “everyday life” refer specifically to bilingual language control?  If bilingual language control (e.g. switching languages) has become automated, then bilinguals are not extensively practicing top-down control and, consequently, should have no advantage compared to monolinguals.  Doesn’t this line of argument predict no differences between the two groups?  On the other hand, even if bilingual language control has become automated, both groups may have extensive everyday experiences in multitasking that hone their general inhibitory control (assuming there is such a thing).  Thus, no matter how I interpret this sentence, I do not understand how it explains the results show effects of anxiety on bilinguals, but not on monolinguals. 

 Second p. 3.  “Testing this hypothesis was not possible given the sample size constraints of this study and we thus encourage a replication involving a larger pool of participants.”  I could not agree more.  But who should bare the burden of investing in a large-scale replication? 

Reviewer 2 Report

This paper investigates the effect of trait anxiety and rumination on inhibitory control, measured by the Anti-saccade task and Simon task, in bilingual and monolingual adults. Although the topic is very interesting and pursues a novel approach by examining the effect of emotional traits on executive control, there are many point that need to be addressed, especially in terms of data analyses. Below are my comments, and I hope that the authors will find them useful to improve their research.

1.      In the current study section, it says “The key question here was whether the commonly postulated inhibitory control differences between monolingual and bilingual individuals, behaviourally and/or on a neural level, would lead to differential effects of trait anxiety and rumination on cognitive performance.” The authors need to clarify what they mean by differential effects and give the readers more specific and detailed hypotheses about their study.

2.      The authors do not provide a rationale for using both pro- and anti-saccade blocks, and why this difference is not considered in the analyses when examining the effect of anxiety traits and rumination on the performance in the antisaccade task.

3.      There are extra lines in Table 3 and 4 that needs to be fixed.

4.      Analyses in section 3.4 and 3.5 should be done in one model. There should be one model each for each dependent variable (i.e., latency of first correct saccade, accuracy, corrective saccades) and the independent variable should be: Group (monolingual, bilingual), Saccade (antisaccade, prosaccade), Trait Anxiety, and Rumination. If there is an interaction between Group and Trait anxiety with accuracy as the dependent variable, this means that the effect of trait anxiety on antisaccade accuracy differs between monolingual and bilingual. Running a model separately for monolingual and bilingual for each predictor (i.e., Trait Anxiety, and Rumination) is very problematic, as running multiple models increases the chance of observing false positives. The same goes for the analyses for the Simon task.

5.      Since reflective pondering and depressive brooding are composites of rumination response, it doesn’t make sense to include them all as predictors as they must highly correlate with one another, as it touches on the multicollinearity issue. I suggest to just use rumination response as a predictor.

6.      I am not sure if the authors can be confident that accuracy on incongruent trials reflect inhibitory control, as traditionally, inhibitory control is measured as the difference between congruent and incongruent trials.

7.      In the discussion it says: The analyses revealed that trait anxiety predicted  lower levels of performance effectiveness only in bilingual participants. Remarkably, this effect was found on all variables of performance effectiveness, including saccadic and behavioural accuracy as well as corrective behaviours. However, when looking at Figure 2a, it seems like the more bilinguals correct their initial response, the greater their trait anxiety is, which is in the opposite direction from others, such as Figure 2a, and Figure 2c. The authors need to acknowledge and clarify this point.

8.      It is unclear why the authors conclude that bilingual advantage in inhibitory control does not affect very early processing effects. (line 450-451)

Reviewer 3 Report

It presents an experiment about the influence of anxiety and rumination on the way how bilingual speakers act in some aspects of language processing (compared to monolinguals). The experiment seems to me well designed and it is well explained.

However, I think that some modifications should be made.  Section 1 (Introduction) is a brief state-ot-the-art concerning the main issue (ACT, anxiety, rumination). It is fine, but, probably, a different organization in subsections would improve the whole presentation. It is odd to find a single subsection 1.1 at the end that consists only of a single paragraph. Besides, where (and which) are the main goals and hypotheses? You can find them in the last section of the paper (Section 4 - Discussion). If goals and hypotheses were clearly presented at the beginning (maybe in the same section 1 or, maybe/preferably in a specific section), the understanding of the paper would be improved. Then, section 4 could be devoted to resume them in light of the results of the experiment and the authors may discuss the actual relevance of these findings for the general current views on this issue (with the pertinent comments on the existing literature) and for future research. I think that most of this is already included, but it is a bit difficult to figure it out. Finally, a conclusions section should also be added (in the shortest version, maybe the current last paragraph modified in a convenient way would serve).   

From a strictly linguistic point of view (my area of expertise), I would like to have more information about the characteristics of the bilingual speakers that took part in the experiment. I mean: languages involved, use of each language in daily life, dominant language, etc. Maybe these factors may have some influence on the results. I'm aware of the difficulties this would imply and I'm also aware that the work is psychologically oriented. It's just a suggestion.

Round 2

Reviewer 1 Report

For bilinguals there is a significant negative correlation between adverse emotional traits and effectiveness (accuracy) in the antisaccade task.  The correlation is not significant for monolinguals.  The first three paragraphs of the discussion are intended to offer an explanation for this pattern, but I do not understand the authors’ account of either one.  I had this difficulty with the original submission, but unfortunately I am still lost.  Maybe it’s me – and accordingly comments from the action editor and other reviewer(s) would be welcome.  

(a) Why do bilinguals show a negative correlation?

(b) Why do monolinguals show no correlation?

(c) Why is there no main effect of language group despite (a) and (b)?

My other remaining comment is rather minor in comparisons and refers to this statement: “Previous research suggests that bilinguals’ level of second language proficiency is a predictor of their cognitive performance (e.g., [11, 76, 77]).” Reference [76] is blank in the reference list.  More important,  we have reported the correlation between L2 proficiency (or the L2/L1 ratio) and several measures of executive functioning (EF) and it never yields a significant correlation.  So previous research does not suggest that L2 proficiency is a consistent predictor of cognitive performance!

(a) Paap & Sawi (2014, Frontiers) & Paap, Johnson, & Sawi (2014, JCP) – 12 measures of EF derived from antisaccade, flanker, Simon & color-shape switching tasks.

(b) Paap, Myuz, Anders, Bockelman, Mikulinsky, & Sawi (2017, JCP) – 3 cued switching tasks (color-shape, letter-digit, animacy-size)

(c) Paap, Anders-Jefferson, Mason, Alvarado, & Zimiga (2018, Frontiers) -   Visual search RT, Proportion Correct, and Slope

(d) Paap, Anders-Jefferson, Mikulinsky, Masuda, & Mason (2019, JML) Composite of interference-scores formed from four nonverbal interference tasks including variants of flanker and Simon.

Author Response

REVIEWER 1 - Second round

For bilinguals there is a significant negative correlation between adverse emotional traits and effectiveness (accuracy) in the antisaccade task.  The correlation is not significant for monolinguals.  The first three paragraphs of the discussion are intended to offer an explanation for this pattern, but I do not understand the authors’ account of either one.  I had this difficulty with the original submission, but unfortunately I am still lost.  Maybe it’s me – and accordingly comments from the action editor and other reviewer(s) would be welcome.  

(a) Why do bilinguals show a negative correlation?

 (b) Why do monolinguals show no correlation?

(c) Why is there no main effect of language group despite (a) and (b)?

Authors’ response: The relationship between adverse emotional traits and effectiveness in the bilingual group was surprising as it is not predicted by the ACT. The interpretation we provided in paragraph 4 of the discussion is one of experience, specifically bilinguals’ lack of experience at modulating effort when performing non-verbal inhibitory control tasks (as argued by the bilingual advantage hypothesis), making effects of anxiety on accuracy more vulnerable. We realise that this argument is highly speculative and, in order to provide evidence for it, a bilingual advantage in inhibition would need be demonstrated alongside the effects reported in this study. Overall, we believe that any interpretation of these findings would require a replication employing a broader range of executive function measures (as stated on page 3.2, lines  483-485). We do struggle making this standpoint any clearer, beyond what is currently discussed in the manuscript.

My other remaining comment is rather minor in comparisons and refers to this statement: “Previous research suggests that bilinguals’ level of second language proficiency is a predictor of their cognitive performance (e.g., [11, 76, 77]).” Reference [76] is blank in the reference list.  More important,  we have reported the correlation between L2 proficiency (or the L2/L1 ratio) and several measures of executive functioning (EF) and it never yields a significant correlation.  So previous research does not suggest that L2 proficiency is a consistent predictor of cognitive performance!

(a) Paap & Sawi (2014, Frontiers) & Paap, Johnson, & Sawi (2014, JCP) – 12 measures of EF derived from antisaccade, flanker, Simon & color-shape switching tasks.

(b) Paap, Myuz, Anders, Bockelman, Mikulinsky, & Sawi (2017, JCP) – 3 cued switching tasks (color-shape, letter-digit, animacy-size)

(c) Paap, Anders-Jefferson, Mason, Alvarado, & Zimiga (2018, Frontiers) -   Visual search RT, Proportion Correct, and Slope

(d) Paap, Anders-Jefferson, Mikulinsky, Masuda, & Mason (2019, JML) Composite of interference-scores formed from four nonverbal interference tasks including variants of flanker and Simon.

Authors’ response: This has been noted on page 3.2 (line 489). The missing reference issue has also been addressed. We are extremely grateful for the reviewers' comments in both round 1 and round 2.

Reviewer 2 Report

I thank the authors for their responses. However, there are still points that I do not understand/agree with the authors, which are outlined below: 

(1)

FIRST ROUND REVIEWER: In the current study section, it says “The key question here was whether the commonly postulated inhibitory control differences between monolingual and bilingual individuals, behaviourally and/or on a neural level, would lead to differential effects of trait anxiety and rumination on cognitive performance.” The authors need to clarify what they mean by differential effects and give the readers more specific and detailed hypotheses about their study.
FIRST ROUND AUTHOR: Thank you for your feedback. In response to comments from yourself as well as the other reviewers, we have made our intentions more clear by including a hypothesis on page 4 (lines 172-175). We do not, however, believe that there is sufficient empirical or theoretical background literature which would justify a directional hypothesis.

SECOND ROUND REVIEWER: I think the key question here is, why would one expect (theoretically) that trait anxiety and rumination influence inhibition in a different manner between monolinguals and bilinguals? As you state in the literature, inhibition abilities (may) differ between bilinguals and monolinguals due to the fact that bilinguals have to juggle two languages in their brain and therefore have more advanced inhibition skills than monolinguals. What I do not understand is why, for example,  trait anxiety or rumination may affect inhibition abilities in bilinguals but not in monolinguals. I believe it is very important to have a theoretical explanation to the hypothesis that the authors lay out. 

(2)

FIRST ROUND REVIEWER:  I am not sure if the authors can be confident that accuracy on incongruent trials reflect inhibitory control, as traditionally, inhibitory control is measured as the difference between congruent and incongruent trials.

FIRST ROUND AUTHOR:  We agree that the way the ACT literature equates performance on difficult cognitive tasks with inhibition calls for scepticism. However, we believe that it is beyond the scope of this paper to discuss this. As noted previously, our decision to operationalise performance on easy and difficult trials as separate dependent variable was driven by the research as well as the theoretical literature surrounding the ACT.

SECOND ROUND REVIEWER: I am not entirely convinced with the author's response that it is beyond the scope of this paper to discuss this issue--I believe it should at least be mentioned in the discussion, given that most researchers in psychology and linguistics regard inhibition measures as the difference between congruent and incongruent trials. In addition, previous research that used incongruent trials as dependent variables in ACT literature should be cited. 

(3)

FIRST ROUND REVIEWER: Analyses in section 3.4 and 3.5 should be done in one model. There should be one model each for each dependent variable (i.e., latency of first correct saccade, accuracy, corrective saccades) and the independent variable should be: Group (monolingual, bilingual), Saccade (antisaccade, prosaccade), Trait Anxiety, and Rumination. If there is an interaction between Group and Trait anxiety with accuracy as the dependent variable, this means that the effect of trait anxiety on antisaccade accuracy differs between monolingual and bilingual. Running a model separately for monolingual and bilingual for each predictor (i.e., Trait Anxiety, and Rumination) is very problematic, as running multiple models increases the chance of observing false positives. The same goes for the analyses for the Simon task.
FIRST ROUND AUTHOR: Whilst we acknowledge that running multiple models may be problematic in some cases, we do not believe that our approach to the data should be adjusted in this manner for the following reasons: 1. Keeping in line with research on the ACT (e.g., Derakshan et al., 2009), which hypothesises that adverse emotions only implicate performance on difficult cognitive tasks, we assessed the relationship of trait anxiety/rumination and performance separately for easy and difficult trials. Once again, we hope that the addition of a hypothesis clarifies why we choose this approach. 2. To the best of our knowledge, this is the first study assessing the effects of both trait anxiety and rumination in one sample. We believe that this is a strength of the study as it serves for a direct comparison of these effects. We do not, however, believe that they should be considered in one model as they correlate highly and significantly in both groups.

SECOND ROUND REVIEWER: Thanks for your detailed explanation. Although I understand your rationale, having so many separate models (i.e., easy vs. difficult, monolinguals vs. bilinguals,. trait anxiety vs. rumination) increases the chance of Type 1 error which is a very big problem when it comes to replicability of the results. Usually, if you run multiple models you should correct for p-value (Bonferroni correction) which seems to not be done in your analyses. I am very concerned about constructing so many regression models, when in actuality, most things could be done in one model. I understand that trait anxiety and rumination highly correlate and should be assessed with model comparisons, but I believe that there is no reason for running separate models between monolinguals and bilinguals. 

(4)

FIRST ROUND REVIEWER: In the discussion it says: The analyses revealed that trait anxiety predicted lower levels of performance effectiveness only in bilingual participants. Remarkably, this effect was found on all variables of performance effectiveness, including saccadic and behavioural accuracy as well as corrective behaviours. However, when looking at Figure 2a, it seems like the more bilinguals correct their initial response, the greater their trait anxiety is, which is in the opposite direction from others, such as Figure 2a, and Figure 2c. The authors need to acknowledge and clarify this point.
FIRST ROUND AUTHOR: Authors’ response: We may need clarification on this feedback - as noted on page 9 (lines 322-323), trials featuring corrective saccades are considered inaccurate and, therefore, it is not surprising that the relationship between adverse emotions and accuracy are in the opposite direction to those between adverse emotions and “inaccuracy

SECOND ROUND REVIEWER:I think I may have misunderstood what the measurement is for "corrective saccades"--was this regarded as "inaccurate" even though they have corrected the initial erroneous saccade? What is the rationale of including this measure, as opposed to the "accuracy" measurement? 

Author Response

Reviewer 2 (second)

I thank the authors for their responses. However, there are still points that I do not understand/agree with the authors, which are outlined below: 

SECOND ROUND REVIEWER: I think the key question here is, why would one expect (theoretically) that trait anxiety and rumination influence inhibition in a different manner between monolinguals and bilinguals? As you state in the literature, inhibition abilities (may) differ between bilinguals and monolinguals due to the fact that bilinguals have to juggle two languages in their brain and therefore have more advanced inhibition skills than monolinguals. What I do not understand is why, for example,  trait anxiety or rumination may affect inhibition abilities in bilinguals but not in monolinguals. I believe it is very important to have a theoretical explanation to the hypothesis that the authors lay out. 

Authors’ response: We have adjusted and expanded upon section 1.1 (page 4, lines 155 - 184) in order to clarify our rationale. To clarify, we did not expect to find no effects in monolinguals, our hypothesis is specifically focused on the between groups comparison. 

SECOND ROUND REVIEWER: I am not entirely convinced with the author's response that it is beyond the scope of this paper to discuss this issue--I believe it should at least be mentioned in the discussion, given that most researchers in psychology and linguistics regard inhibition measures as the difference between congruent and incongruent trials. In addition, previous research that used incongruent trials as dependent variables in ACT literature should be cited. 

Authors’ response: We have added a brief discussion of this issue on pages 3.2 and 4.2 (lines 518-525).

SECOND ROUND REVIEWER: Thanks for your detailed explanation. Although I understand your rationale, having so many separate models (i.e., easy vs. difficult, monolinguals vs. bilinguals,. trait anxiety vs. rumination) increases the chance of Type 1 error which is a very big problem when it comes to replicability of the results. Usually, if you run multiple models you should correct for p-value (Bonferroni correction) which seems to not be done in your analyses. I am very concerned about constructing so many regression models, when in actuality, most things could be done in one model. I understand that trait anxiety and rumination highly correlate and should be assessed with model comparisons, but I believe that there is no reason for running separate models between monolinguals and bilinguals. 

Authors’ response: Ideally, the current study would have employed a 2x2x2 design, with anxiety group being one of the independent variables. However, our sample size did not allow for a tertile split of each group, making an alignment with the analyses used in previous studies not possible. We have acknowledged this limitation in our discussion and very much agree that this study requires a larger-scale replication. However, we believe that treating trait anxiety and rumination as continuous and comparing regression coefficients in the manner we did was the most appropriate approach, especially given that the ACT literature rarely evaluates distractor cost as a dependent variable, as clarified on pages 3.2 and 4.2 (lines 518-525). Furthermore, we did not employ a correction as the comparisons of the coefficients was the main aim of our analysis. Overall, we do not believe that making changes to our analysis approach will add to the quality of this manuscript; however, we will provide the raw data upon publication in order to allow for further analyses to be conducted by interested parties.

SECOND ROUND REVIEWER:I think I may have misunderstood what the measurement is for "corrective saccades"--was this regarded as "inaccurate" even though they have corrected the initial erroneous saccade? What is the rationale of including this measure, as opposed to the "accuracy" measurement? 

Authors’ response: Corrective saccades were classified as inaccurate which was informed by the literature, as detailed in the introduction (page 3, lines 111-114).